# The Use of *Opuntia ficus-indica* Mucilage and *Aloe arborescens* as Edible Coatings to Improve the Physical, Chemical, and Microbiological Properties of 'Hayward' Kiwifruit Slices

Giuseppe Sortino, Paolo Inglese, Vittorio Farina, Roberta Passafiume and Alessio Allegra *

Department of Agricultural, Food and Forest Sciences, Università degli Studi di Palermo, Edificio 4, Ingresso H, 90128 Palermo, Italy; giuseppe.sortino@unipa.it (G.S.); paolo.inglese@unipa.it (P.I.); vittorio.farina@unipa.it (V.F.); roberta.passafiume@unipa.it (R.P.)
* Correspondence: alessio.allegra@unipa.it; Tel.: +39-09123861234

**Abstract:** Edible coatings (EC) with mucilage of *Opuntia ficus-indica* or *Aloe arborescens* are promising to extend the shelf life of fresh-cut fruit products by reducing weight loss and microbial spoilage. In this work, fresh-cut kiwifruits (cv. Hayward) were coated in solutions with mucilage extracted from *Opuntia ficus-indica* (MC) and *A. arborescens* (AL). We used three alternative treatments with AL or MC, MC+AL, and with distilled water as control, and stored in passive atmosphere at 5 °C for 3, 5, 7, and 9 d, respectively. For all treatments at each storage period, firmness, weight loss, color, visual quality score, respiration rate, pectin content, and microbiological characteristics were observed. The treatments with mucilage and *A. arborescens* applied on fresh-cut kiwi slices showed different significant effects until 9 days of storage, in terms of firmness and total pectin. Microbial spoilage analysis revealed the beneficial effects of this strategy after 3 d, particularly in terms of bacteria and yeast. *A. arborescens* provided a reduction of microbial activity, probably due to the higher quantity of aloin if related to Aloe species. Furthermore, the treatment with MC and AL increased the total pectin content, showing positive effects in terms weight loss and firmness. The results showed that the MC+AL treatment improved the visual score of fresh-cut kiwi fruit until 7 d of storage.

**Keywords:** *Actinidia deliciosa*; pectin contents; fresh-cut; post-harvest; bio-based films; visual color

## 1. Introduction

Kiwifruit (*Actinidia* spp.) is a typical climacteric fruit that, after harvest, goes through three distinct softening phases that are temporally well separated. Pectin retained in the cell wall starts to "soften" during ripening; this process clearly precedes both pectin solubilization and depolymerization (Phase 1) [1].

The degradation of solubilized pectin and loss of middle lamellae in kiwifruit (Phase 2) are processes initiated in the second softening phase, with a peak in the last softening phase, where cell wall disintegration is completed (Phase 3) [1]. After harvest, kiwifruits can be stored for a long period and sold for fresh consumption or for processing as fresh-cut. The peeling and slicing involved in minimal fruit processing can cause physical damages, and an increase of polyphenoxidase and peroxidase begins browning in the flesh. In fresh-cut fruit, the rapid softening and deterioration likely involve membrane and cell wall catabolism, accelerated or otherwise altered in response to physical wounding [2].

In recent years, limited research works have been carried out on applying various kinds of natural edible coatings to extend fresh-cut kiwifruit during storage. The use of edible coatings could be an efficient alternative to the chemical treatments applied to preserve fresh-cut kiwifruit. *Opuntia ficus-indica* (OFI) mucilage, *A. vera*, and *A. arborescen* coatings reduce physiological disorders and gas exchange [3,4]. Moreover, these coatings can maintain total soluble solids, a higher concentration of total phenols and ascorbic acid, and a better antioxidant activity when compared to the control [5–7]. The most

studied gel from any *Aloe* spp. has been *A. vera* or *A. arborescens*, where the gel is found in a clear internal zone located between the abaxial and adaxial mesophyll. The chemical composition of the aloe species gel is very complex, composed mainly of polysaccarides and soluble sugars, followed by proteins; most of these are enzymes, amino acids, and vitamins [8]. Zapata et al. [9] showed that *A. arborescens* gel had more aloin concentration, total phenolics, total antioxidant activity (hydrophilic and lipophilic fractions), and a low level of putrescine and spermidine than *A. vera* gel content. However, the use of *A. arborescens* showed overall inhibitions of fungal growth on inoculations in vitro test while the use of OFI mucilage did not show a positive effect on concentration of pseudomonads and yeasts during storage of breba fig fruit [10]. OFI mucilage is present in cladodes, in the Golgi apparatus, and the mucilage synthesis probably takes place in it and in the vesicles derived from it [11]. These polysaccharides swell when dissolved in water, or in some cases, form colloidal and very viscous suspensions or jellied masses [12]. In particular, OFI mucilage has complex polymeric substances of a carbohydrate nature; a highly branched structure [13]; and is composed of arabinose, galactose, rhamnose, xylose and galacturonic acid [14]. The use of a natural edible coating could replace calcium-based treatments and compounds made from different fermentation or polymerization reactions (e.g., HPMC, Xanthan gum, etc.). OFI mucilage and *A. vera* coatings have been previously studied by Allegra [14] and Benitez [3], regarding the maintenance of quality in kiwifruit slices during storage. However, those studies showed that non-combined treatments have not improved the physical and microbiological qualities of fresh-cut kiwi during storage. Benitez showed that, with kiwi slices, coated and uncoated samples were given similar scores during storage, except in the bitterness attributes.

The objective of the present work was to evaluate the efficacy of *O. ficus-indica* coatings with *A. arborescens* (2%) in prolonging the shelf life of kiwifruit slices. We focused our study on the effects of OFI mucilage with *A. arborescens* on the visual quality score, firmness, and microbial load of minimally processed kiwifruit slices, during 9 days of storage.

## 2. Materials and Methods

### 2.1. Fruits and Preparation of the Edible Coating

'Hayward' kiwifruits (*Actinia deliciosa*) were purchased from a supermarket belonging to a large, very well-known organized distributor in Italy, which provides medium-high quality products (Conad, Palermo, Italy). Kiwifruits were selected based on their firmness (49 N $\pm$ 2.2) and soluble solid total (12.5 °Brix $\pm$ 0.2) stored in air at 5 $\pm$ 0.5 °C (RH = 85%) for 24 h. After storage, the fruits were dipped in chlorinated water (100 ppm of free chlorine) for 6 min. Damaged fruits (bruised or showing other physical decays) were removed, and a total of 400 fruits with an average firmness of 48.2 $\pm$ 3.5 N and weight of 100 $\pm$ 4.2 g were processed.

Cactus pear (*O. ficus-indica* (L.) Mill.) cladodes were collected from an experimental field near Roccapalumba (37°48′ N; 13°38′ E; 350 m asl), western Sicily, Italy. Promptly after harvest, cladodes were transported to the post-harvest laboratory of the University of Palermo and were cut and cubed (2 cm$^3$). Samples were homogenized (20% $w/v$) in distilled water with water ratio 1:1.5. The solution was maintained at 40 °C for 90 min and centrifuged (model CS6R) at 3000 rpm $\times$ 20 min. The supernatant was boiled to halve the initial volume, and ethanol at 99% was added in ratio 1:2, in order to reduce the amount of alcohol used in the precipitation [15]. Afterwards, the solution was stored at 4 $\pm$ 1 °C for 48 h to allow a better aggregation of the mucilage. The last phase involved the elimination of the supernatant and the soaking of the pure mucilage.

Matured leaves of the *A. arborescens* plant were harvested at the experimental field of the University of Palermo (Italy). *A. arborescens* gel was prepared from 1 kg of leaves taken from 10-year-old plants. The leaves were cleaned externally with a knife, removing the margin, and were then cut lengthwise. The parenchyma (from which the gel is obtained) was separated from the epidermis. The gelatinous parenchyma was homogenized with Ultra-Turrax (Ultra-Turax T25, Janke and Kunkle, IKa Labortechnik, Breisgau, Germany)

for 5 min at 24,500 rpm, thus obtaining a mucilaginous gel, and was subsequently filtered to eliminate the fibrous portion [16].

### 2.2. Fruit Processing

Kiwifruits were peeled manually and cut into slices with a semiautomatic machine. Slices were $2.2 \pm 0.3$ cm thick and $5.4 \pm 1.1$ cm diameter, characterized by the Hunter's parameters L* = $66.6 \pm 2.4$, a* = $7.8 \pm 1.3$, and b* = $25.4 \pm 2.1$, with a total solid soluble content (SST) of $12.9 \pm 2.2$ °Brix and titratable acidity (TA) of $1.5 \pm 0.4$ (g L$^{-1}$). Fresh-cut slices were dipped in the coating solution for 60 s; the excess coating was drained, and the coated slices were dried in a forced-air dryer (20 °C) for 10 min.

The coating treatment consisted of: (a) 30 g of pure mucilage extract, 500 mL distilled water with 2% of *A. arborescens* and 50 mL glycerol as a plasticizer (MC+AL); (b) 30 g of mucilage extract, 500 mL distilled water, 50 mL glycerol (MC); (c) 500 mL distilled water with 2% of *A. arborescens* and 50 mL glycerol as a plasticizer. The control trial (CTR) was set up by sliced fruits dipped in distilled water. Approximately 100 g $\pm$ 0.8 of kiwifruit slices were packed in polyethylene terephthalate (PET) packages and sealed with a composite film (PP-PET, 64 μm, O$_2$ permeability = $5.30 \times 10^{-8}$ μL m$^{-2}$ s$^{-1}$ Pa$^{-1}$). Packages were stored at 5 °C and 90% relative humidity (RH) for 9 d. Physico-chemical and microbiological quality parameters were analyzed on six slices used as single replicates (4 treatments $\times$ 5 time of storage $\times$ 6 replicates = 120 box), at the beginning of the experiment (after coating/dipping = day 0) and at 3, 5, 7, and 9 d after storage.

### 2.3. Firmness

Firmness was evaluated by a puncture test on kiwifruit slice flesh using a TA-XT Plus texture analyzer (Stable Micro Systems, Surrey, England). Firmness measurements were taken as the medium force value obtained during the test by a stainless steel probe with 4 mm diameter penetrating inside the fruit for 4 mm, at 1 mm/s. Average values were calculated from the results of at least six measurements in different slices for each sample. Measures were taken in the fruit outer pericarp (green flesh), where the fast rate of softening compromised fruit quality.

### 2.4. Weight Loss (%)

The weight of individual bags was recorded immediately after the treatment (day 0) and at the different sampling times (3, 5, 7, and 9 d during storage). Weight loss was expressed as the percentage reduction with respect to initial time, using the following equation:

$$\% \text{ Weight loss} = [(\text{Initial fruit bags weight} - \text{Final fruit bags weight}) \times 100]/\text{Initial fruit bags weight}$$

### 2.5. Total Soluble Solids Content and Titratable Acidity

Soluble solids total concentration was determined with a hand-held refractometer, and pH was determined by a pH meter. Titratable acidity (expressed as % citric acid) was determined by titration of 10 mL of juice with 0.1 M NaOH to an endpoint of pH 8.1 [3].

### 2.6. Visual Appearance Score

To measure the effect of cold storage on kiwifruit sensory traits at each storage time (0, 3, 5, 7, and 9 d), six slices, used as single replicates for treatment (MC+AL, MC, AL, and CTR), were scored by each of a six-judge trained panel, who generated a list of descriptors in a few preliminary meetings. All panelists participated in 10 d of training and 6 d of tasting, lasting for a total of 4 weeks. Visual appearance was focused on color, visible structural integrity, and visual appearance [17]. The different descriptors were quantified using a subjective 5 pt. scoring scale with 5 = very good, 4 = good, 3 = sufficient (limit of marketability), 2 = poor (limit of usability), and 1 = very poor (inedible).

### 2.7. Package $O_2$ and $CO_2$ Analysis

$CO_2$ and $O_2$ levels (Kpa) were measured on each package at the beginning of each experiment and after 3, 5, 7, and 9 d of storage, using a PBI Dansensor Checkpoint $O_2$ and $CO_2$ analyzer (Topac, Hingham, MS, USA) with zirconium and infrared detectors, respectively.

### 2.8. Color

The superficial color of the kiwifruit slices was determined using a Minolta colorimeter CR-400 model (Minolta Camera, Osaka, Japan) in the CIElab space. The L* (lightness), a* (greenness (−) to redness (+)), and b* (blueness (−) to yellowness (+)) values were recorded for each treatment at 3, 5, 7, and 9 d [5]. Total color difference (ΔE*) expressed the magnitude of the difference between the initial non-aged color pulp (zero time) and storage-aged samples. Total color difference (ΔE*) was calculated according to the following:

$$\Delta E^* = [(\Delta L^*)^2 + (\Delta a^*)^2 + (\Delta b^*)^2]^{1/2} \tag{1}$$

where $\Delta a^* = a^* - a_0^*$, $\Delta b^* = b^* - b_0^*$, $\Delta L^* = L^* - L_0^*$; $a_0^*$, $b_0^*$, and $L_0^*$ are the corresponding blank values of control sample, and $a_0^* = -2.00$, $b_0^* = 4.00$, and $L_0^* = 80.00$, respectively [5]. All data were reported as mean ± standard error (SE, $n = 6$).

### 2.9. Microbiological Analysis

Fruit samples and mucilage were microbiologically investigated for total mesophilic microorganisms (TMM) and the undesired (spoilage and/or pathogenic) microbial groups. The fruits (25 g), *A. arborescens* (10 mL), and mucilage (10 mL) were suspended in Ringer's solution (Sigma-Aldrich, Milan, Italy) with a ratio 1:10 (fruit:diluent), homogenized for 2 min at the highest speed with a stomacher (BagMixer® 400, Interscience, Saint Nom, France), and serially diluted. The cell suspensions were inoculated as follows: TMM on plate count agar (PCA), incubated at 30 °C for 72 h; Pseudomonas on Pseudomonas agar base (PAB), supplemented with 10 mg/mL cetrimide fucidin, incubated at 20 °C for 48 h; yeasts on yeast potato dextrose (YPD) agar, incubated at 25 °C for 48 h. All materials and supplements were purchased from Oxoid (Milan, Italy). Count plates were carried out in duplicate for each trial.

### 2.10. Pectin Analysis

The pectic substances from fruits were extracted according to the methods described by Rouse and Atkins [18,19]. Frozen 'Hayward' *A. deliciosa* were thawed at 4 °C for 24 h; approximately 20 g were weighed into a plastic cup and homogenized at low speed with an Ultra-Turrax tissue mixer for 2 min. The tissue sample (5 g) was weighed into a 50 mL round-bottom plastic centrifuge tube. Hot 100% ethanol (30 mL) was added to the tubes. The samples were stirred thoroughly with a glass rod, heated in a boiling water bath for 10 min, cooled, and centrifuged at 10,000 rpm for 10 min. The alcoholic supernatant was decanted and discarded. The precipitate was extracted with 30 mL 100% ethanol, centrifuged, and the supernatant was discarded. The residue was transferred from the centrifuge tube to a 57 mm aluminum weighing dish (Fisher Scientific Inc. Italia, Milan, Italy). Samples were dried for 24 h in a conventional oven at 35 °C, weighed, and ground in a 65 mortar with a pestle. The precipitate from the alcohol solution was designated as alcohol-insoluble solids (AIS).

#### 2.10.1. Water Soluble Pectin

Dried alcohol-insoluble solid (AlS) (80 mg) was weighed into a 50 mL centrifuge tube. Distilled water (20 mL) was added, and samples were stirred with a glass rod for 1 min. Samples were centrifuged at 10,000 rpm for 10 min and filtered through Whatman No. 1 filter paper to obtain water-soluble pectin (WSP). The extraction procedure was repeated once. The supernatants were collected and combined in a 100 mL volumetric flask. Distilled water was added to dilute the tube extract solution to volume.

### 2.10.2. Oxalate Soluble Pectin

The residue was dispersed in 20 mL of an aqueous solution containing 0.25% ammonium oxalate and 0.25% oxalic acid and stirred with a glass rod for 1 min. Samples were refluxed in a boiling water bath for l h, centrifuged at 10,000 rpm for 10 min, and filtered through Whatman No. 1 filter paper to obtain oxalate-soluble pectin (OSP). The oxalate extraction was repeated once. Supernatants were collected and diluted to 100 mL with distilled water in a volumetric flask.

### 2.10.3. Total Pectin

Total pectin (TP) was extracted following the method of Ahmed and Labavitch [20]. Dried AIS (5 mg) was weighed into a 30 mL beaker containing a magnetic stir bar. Concentrated sulfuric acid (2 mL) was added to the beaker, and the mixture was swirled gently. The beaker was placed on a stir plate and stirred gently, and 0.5 mL of distilled water were added dropwise. Stirring continued for 5 min; an additional aliquot of 0.5 mL distilled water was added dropwise. Stirring continued further for approximately 30 min until the dissolution of the AIS was complete. The dissolved sample was filtered through glass wool into a 25 mL volumetric flask. Each beaker was rinsed several times with distilled water, combined in a 25 mL flask, and diluted to volume. The solution was filtered through glass wool once more before use.

### 2.10.4. Non-Extractable Pectin

The difference between TP and the sum of WSP and OSP was used to determine the amount of non-extractable pectin (NXP), which was the protopectin fraction. TP, WSP, and OSP extractions were completed in duplicate for all samples. All pectin extracts were stored at 5 °C for 12 h before analysis.

### 2.10.5. Pectin Content Determination

Pectin extracts contents were analyzed by the m-hydroxydiphenyl method [21]. The extract from each sample (1 mL) was pipetted into a 16 × 150 mm test tube. Sulfuric acid tetraborate solution (0.0125 M sodium tetraborate in concentrated sulphuric acid) (6 mL) was added to each of the tubes in an ice water bath and mixed carefully using a Vortex mixer at moderate speed, with intermittent stopping to assure complete mixing. Duplicate samples were prepared for each pectin measurement with a corresponding blank. Tubes were heated in a boiling water bath for 5 min and immediately placed in ice water to cool. To duplicate tubes, 0.1 mL aliquot of 0.15% m-hydroxydiphenyl, was added to develop color. To the blank tube, 0.1 mL 0.5% sodium hydroxide was added. All samples and blanks were mixed using the Vortex mixer and allowed to stand for 15 min at room temperature. The absorbance of the samples following chromogen formation was measured at the wavelength of 520 nm using a Spectrophotometer. Galacturonic acid was used as a standard. A solution consisting of 1 mL distilled water, 6 mL sulfuric acid/tetraborate, and 0.1 mL 0.5% sodium hydroxide was used as reagent blank. The determination of the pectin was carried out according to Yu et al. [22].

### 2.11. Statistical Analysis

The experimental design consisted of three coating treatments and the untreated control, with observations made at 0, 3, 5, 7, and 9 d after coating. Analysis of Variance (ANOVA) to compare collected data was used (Systat 13.0 (Systat Software, Inc. San Jose, CA USA) was used as statistical software). Significant differences ($p \leq 0.05$) were evaluated using the Tukey's test.

## 3. Results and Discussion

### 3.1. Solid Soluble Total (SST), Titratable Acidity (TA), Color and Visual Score

Kiwifruit slice SST did not change significantly between treatments during storage time, but after 3 d, the sample slices treated with mucilage (MC) showed significant

differences compared to the other treatments (CTR, AL, MC+AL). The control slices (CTR) and MC treatment showed an increase of mean values from 12.9 at 14.1 °Brix and 12.9 at 13.9 °Brix, respectively (Table 1). This is related to the sharp increase of ethylene and respiration rate [23]. Jordan et al. [24] observed that most of the carbohydrates in kiwifruit at harvest were starch, which was hydrolyzed into sugar during further ripening and may lead to the increase of TSS. Untreated kiwifruit slices showed a sharp, significant ($p \leq 0.05$) increase in SST, while no significant differences were observed in "MC+AL" and AL treatments during storage time (Table 1). Similar results were obtained by Sicari et al. [25] during conservation of strawberry fruits coated with *A. arborescens* gel. A reverse trend during storage time was observed (Table 2) in terms of values of TA; CTR and AL treatments both showed a significant decrease ($p \leq 0.05$) from cutting at day 9, while slices coated with MC+AL and MC showed no significant decrease during storage time. The reduction of acidity content in kiwifruit is related to softening and therefore may be associated with the consumption of organic acids under the mechanism of respiration and gluconeogenesis. Generally, the titratable acid content of most fruits increases or remain unchanged during the ripening process, and decreases only at a very ripe stage [26]. The treatment with MC+AL and MC influenced TA content by maintaining the mean values between 1.6 and 1.4%. In breba fig fruit, the malic acid, glutamic acid, citric acid, pipecolic acid, benzoic acid, malonic acid, pelargonic acid, stearic acid, and linoleic acid were higher in sample coated with OFI mucilage than in control samples analyzed at commercial harvest time [27].

**Table 1.** Solid soluble total of fruit slices of *Actinidia deliciosa* (Liang. Ferguson) coated with *O. ficus-indica* (OFI) mucilage (MC), or mucilage + *A. arborescens* (MC+AL) or *A. arborescens* (AL) not treated (CTR), just after being coated (0) and at 3, 5, 7, 9 d of storage at 5 °C. Different lowercase letters indicate significant differences between treatments at each sampling date. Different capital letters indicate significant differences between sampling date for the same treatment. Tukeys' significant test was applied at $p \leq 0.05$. Data are means ± S.E. ($n = 3$).

| Time of Storage (Days) | Solid Soluble Total (°Brix) | | | |
|---|---|---|---|---|
| | **CTR** | **MC** | **MC+AL** | **AL** |
| 0 days | 12.9 ± 0.2 A nsA | 12.9 ± 0.2 NS | 12.9 ± 0.2 A | 12.9 ± 0.2 NS |
| 3 days | 13.2 ± 0.2 nsA | 13.1 ± 0.1 | 13.3 ± 0.2 A | 13.0 ± 0.3 |
| 5 days | 13.5 ± 0.2 aA | 13.0 ± 0.1 b | 13.5 ± 0.4 aA | 13.2 ± 0.4 a |
| 7 days | 13.7 ± 0.4 nsA | 13.3 ± 0.3 | 13.5 ± 0.2 A | 13.3 ± 0.2 |
| 9 days | 14.1 ± 0.1 nsAB | 13.9 ± 0.4 | 13.6 ± 0.5 AB | 13.5 ± 0.2 |

**Table 2.** Titratable acidity of fruit slices of *Actinidia deliciosa* (Liang. Ferguson) coated with *O. ficus-indica* (OFI) mucilage (MC), or mucilage + *A. arborescens* (MC+AL) or *A. arborescens* (AL) not treated (CTR), just after being coated (0) and at 3, 5, 7, 9 d of storage at 5 °C. Different lowercase letters indicate significant differences between treatments at each sampling date. Different capital letters indicate significant differences between sampling date for the same treatment. Tukeys' significant test was applied at $p \leq 0.05$. Data are means ± S.E. ($n = 3$).

| Time of Storage (Days) | Titratable Acidity (g L$^{-1}$) | | | |
|---|---|---|---|---|
| | **CTR** | **MC** | **MC+AL** | **AL** |
| 0 days | 1.8 ± 0.2 nsA | 1.8 ± 0.2 NS | 1.8 ± 0.2 NS | 1.8 ± 0.2 A |
| 3 days | 1.5 ± 0.2 aA | 1.7 ± 0.3 a | 1.6 ± 0.2 b | 1.7 ± 0.5 bA |
| 5 days | 1.3 ± 0.1 aAB | 1.6 ± 0.2 b | 1.6 ± 0.1 b | 1.6 ± 0.3 bA |
| 7 days | 1.2 ± 0.3 aAB | 1.5 ± 0.1 b | 1.6 ± 0.4 b | 1.5 ± 0.2 bA |
| 9 days | 0.9 ± 0.6 B | 1.4 ± 0.5 b | 1.4 ± 0.2 b | 1.0 ± 0.3 aB |

The brightness values (L*) showed a significant difference between treatments at all storage time points (Table 3). After cutting, MC+AL samples showed a slow decrease

from the 3rd to the 7th day, while at the 9th day there was a significant decrease of 30% of brightness. The L* value of untreated kiwifruit slices showed an increase from the 5th to the 7th day: 58.4 and 70.4, respectively. The development of translucency, as a sign of tissue injury, could be triggered by an excessive in-package accumulation of $CO_2$ [28] or by water loss from the inside to the outside of cut fruit.

**Table 3.** Color L* of fruit slices of *Actinidia deliciosa* (Liang. Ferguson) coated with *O. ficus-indica* (OFI) mucilage (MC), or mucilage + *A. arborescens* (MC+AL) or *A. arborescens* (AL) not treated (CTR), just after being coated (0) and at 3, 5, 7, 9 d of storage at 5 °C. Different lowercase letters indicate significant differences between treatments at each sampling date. Different capital letters indicate significant differences between sampling date for the same treatment. Tukeys' significant test was applied at $p \leq 0.05$. Data are means ± S.E. ($n = 3$).

| Time of Storage (Days) | L* (CieLab) | | | |
|---|---|---|---|---|
| | **CTR** | **MC** | **MC+AL** | **AL** |
| 0 days | 66.6 ± 2.4 nsA | 66.6 ± 2.4 nsA | 66.6 ± 2.4 nsA | 66.6 ± 2.4 nsA |
| 3 days | 60.3 ± 1.9 cB | 67.4 ± 1.9 abA | 68.4 ± 5.3 aA | 68.2 ± 2.1 aA |
| 5 days | 58.4 ± 1.9 dB | 60.1 ± 2.9 cB | 67.4 ± 3.2 bA | 69.9 ± 2.0 aA |
| 7 days | 70.4 ± 2.8 aC | 58.9 ± 3.4 cC | 67.1 ± 1.9 bA | 51.4 ± 5.8 dB |
| 9 days | 51.5 ± 1.9 bD | 55.2 ± 5.3 aD | 45.7 ± 3.7 cB | 52.1 ± 4.7 bB |

The color change measured with ΔE in slices coated with mucilage + *A. arborescens* (MC+AL) showed no relevant differences during the first 3 d of storage compared to other treatments, while after 3 d the mean values of ΔE showed significant differences between CTR, MC, and AL treatments (Table 4). During storage, no significant ($p \leq 0.05$) changes in slices coated with MC+AL were observed; in fact, only after 9 d was a change loss of 3.5% seen. A change of color occurred in untreated slices (CTR), showing significant difference between MC and AL.

**Table 4.** ΔE% of fruit slices of *Actinidia deliciosa* (Liang. Ferguson) coated with *O. ficus-indica* (OFI) mucilage (MC), or mucilage + *A. arborescens* (MC+AL) or *A. arborescens* (AL) not treated (CTR), just after being coated (0) and at 3, 5, 7, 9 days of storage at 5 °C. Different lowercase letters indicate significant differences between treatments at each sampling date. Different capital letters indicate significant differences between sampling date for the same treatment. Tukeys' significant test was applied at $p \leq 0.05$. Data are means ± S.E. ($n = 3$).

| Time of Storage (Days) | ΔE (%) | | | |
|---|---|---|---|---|
| | **CTR** | **MC** | **MC+AL** | **AL** |
| 3 days | 1.2 ± 0.3 nsA | 0.8 ± 0.2 A | 0.7 ± 0.2 A | 1.3 ± 0.5 A |
| 5 days | 3.1 ± 0.4 aB | 1.2 ± 0.4 bA | 0.9 ± 0.1 cA | 1.8 ± 0.4 bA |
| 7 days | 4.7 ± 0.8 aB | 2.2 ± 0.3 bAB | 1.1 ± 0 cA | 2.1 ± 0.4 bA |
| 9 days | 7.9 ± 1.9 aC | 5.2 ± 1.1 bB | 3.3 ± 0.9 cB | 5.1 ± 0.7 bB |

Kiwifruit slice control showed a continuous and significant ΔE increase throughout the storage period ($p \leq 0.05$) and, at the end of the storage period, showed higher color loss between the coated slices (Table 4). Because a ΔE value > 4 is considered perceptible to human eyes (CIE 2004), all differences, between the color at cutting time and the measured one during storage, were visible only after 7 d of storage.

MC+AL and MC treatments had the best visual appearance scores for all sampling times, while CTR and AL treatments had the fastest decay rates, becoming almost non-marketable, respectively, after 3 and 5 d of storage (Figure 1). In term of visual score, kiwifruit slices coated with MC+AL were still marketable after 7 d of storage, with a rapid decline of values similar to MC treatment.

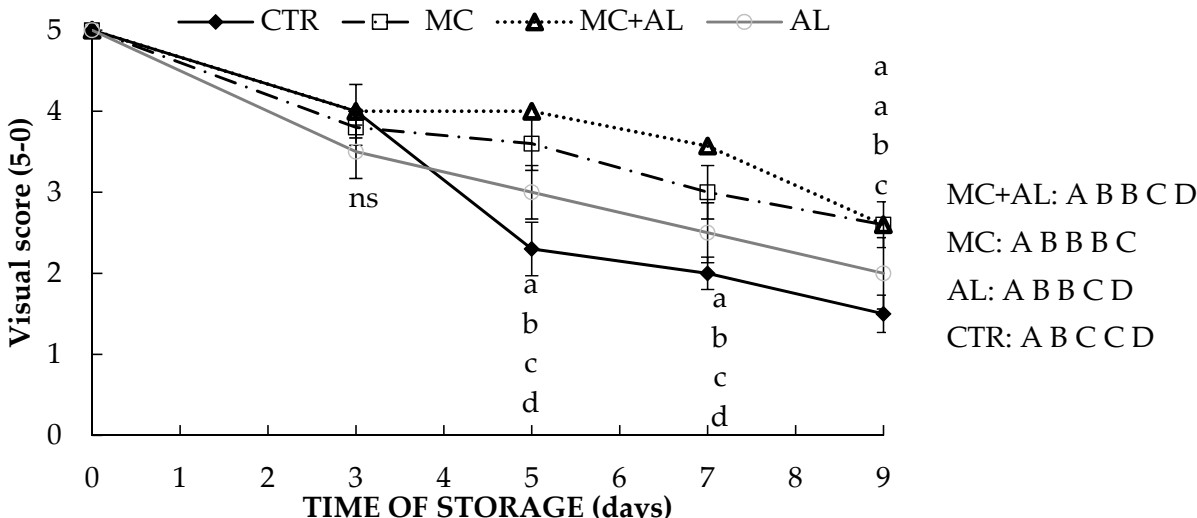

**Figure 1.** Visual score of fruit slices of *Actinidia deliciosa* (Liang, Ferguson) coated with *O. ficus-indica* (OFI) mucilage (MC), or mucilage + *A. arborescens* (MC+AL) or *A. arborescens* (AL) not treated (CTR), just after being coated (0) and at 3, 5, 7, 9 days of storage at 5 °C. Different lowercase letters indicate significant differences between treatments at each sampling date. Different capital letters indicate significant differences between sampling date for the same treatment. Tukeys' significant test was applied at $p \leq 0.05$. Data are means ± S.E. ($n = 3$).

Differences between treatments appeared 3 d after storage when untreated; CTR slices had the lowest value, while at the same time, no differences occurred between MC+AL and MC coated slices (Figure 1).

### 3.2. Firmness, Weight Loss and Pectin Content

Fruit softening is initiated and sustained by developmental and hormonal signals, and at the biochemical level it involves extensive remodeling and breakdown of the cell wall structure [29] with solubilization and depolymerisation of pectins [30]. The polysaccharide composition of *A. vera* gel [31] and OFI mucilage [14] has proved to be highly effective as a moisture barrier.

In our work, MC+AL kiwifruit slices had the highest firmness throughout the storage period (Figure 2). These results can be linked with the pectin total content detected in MC+AL samples. Indeed, fresh-cut processing causes the disruption of surface cells and injury stress of underlying tissues. The increases of pectin with OFI mucilage and *A. arborescens* coating could reduce the increase of membrane permeability and tissue disturbance. Moreover, calcium content into OFI mucilage [32] can maintain cell wall stability and integrity, as well as determining fruit quality [33]. Stabilization of membrane systems and formation of calcium pectates increased the rigidity of the middle lamella and cell walls and retarded polygalacturonase (PG) activity [34]. Composite coatings of polysaccharides could increase the lipid content, which, as a result, could further reduce the weight loss and firmness [35].

During the first 3 d of storage, there occurred a reduction of firmness from 48.2 N to 40–37 N in all treatments, and no significant differences were observed. After 5 d, kiwifruit slices MC+AL and MC treatments showed significant differences ($p \leq 0.05$) compared to untreated slices (CTR) and AL treatment. However, on the 7th day, the values of sample slices treated with MC+AL rapidly declined to values similar to CTR and AL treatment. At the last sampling time (Figure 2), significant differences occurred between MC and CTR, AL, and MC+AL treatments, showing, respectively, a firmness loss of 40.2, 54.8, 53.7, and 50.1% compared to firmness measured after cutting (48.2 N).

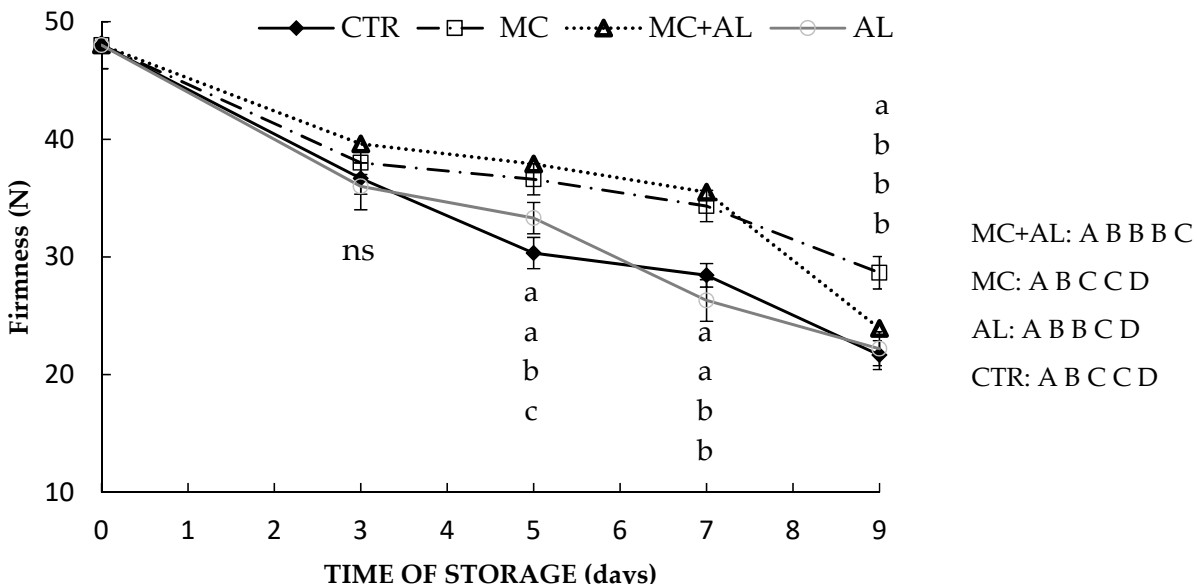

**Figure 2.** Firmness of fruit slices of *Actinidia deliciosa* (Liang, Ferguson) coated with *O. ficus-indica* (OFI) mucilage (MC), or mucilage + *A. arborescens* (MC+AL) or *A. arborescens* (AL) not treated (CTR), just after being coated (0) and at 3, 5, 7, 9 days of storage at 5 °C. Different lowercase letters indicate significant differences between treatments at each sampling date. Different capital letters indicate significant differences between sampling date for the same treatment. Tukeys' significant test was applied at $p \leq 0.05$. Data are means ± S.E. ($n$ = 3).

After 3 d, weight loss was higher in untreated than in AL, MC, and MC+AL kiwifruit slices, while after the cutting of kiwifruit, no significant differences were observed between treatments (Figure 3).

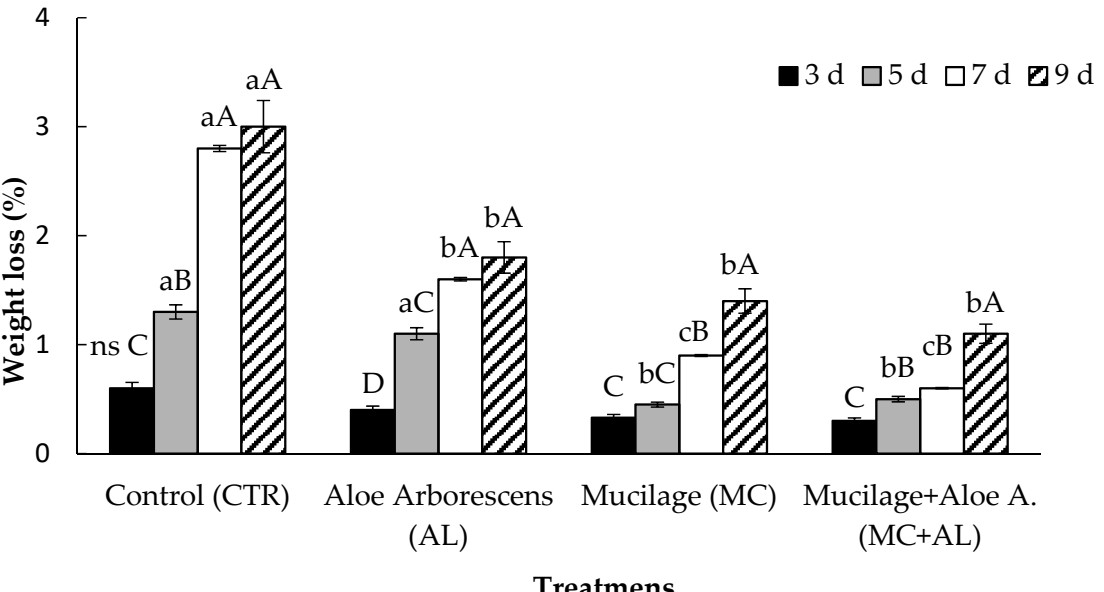

**Figure 3.** Weight loss (%) of fruit slices of *Actinidia deliciosa* (Liang, Ferguson) coated with *O. ficus-indica* (OFI) mucilage (MC), or OFI mucilage (MC), or mucilage + *A. arborescens* (MC+AL) or *A. arborescens* (AL) not treated (CTR), just after being coated (0) and at 3, 5, 7, 9 days of storage at 5 °C. Different lowercase letters indicate significant differences between treatments at each sampling date. Different capital letters indicate significant differences between sampling date for the same treatment. Tukeys' significant test was applied at $p \leq 0.05$. Data are means ± S.E. ($n$ = 3).

The weight loss of CTR and AL samples slices significantly increased ($p \leq 0.05$) after 5 d of storage; at the last sampling time, no weight loss differences occurred between CTR and AL kiwifruit treatments (Figure 3).

In kiwifruit slices coated with MC or MC+AL, the weight loss (%) showed a significant difference ($p \leq 0.05$) compared to AL treatment and CTR slices. At the 7th and 9th day of storage, MC and MC+AL treatments showed a percentage increase of 0.90% and 0.60% and 1.40% and 1.20%, respectively. At the same time points, CTR and AL showed a percentage increase of 2.8% and 3.0% and 1.60% and 1.80%, respectively. In our work, we used the hydrophilic character of the OFI mucilage coating as a barrier to water transfer, retarding dehydration and, therefore, prolonging the firmness during fruit shelf life, while the use of aloe coating at 2% did not show effects on water loss fruit. There is also evidence for strong covalent interactions between pectin populations and cellulose [36,37]. The addition of *A. arborescens* (2%) to mucilage solution contributed to a significant effect on total pectin (Figure 4) on the 3th day of storage, but not during storage, compared to the MC treatment.

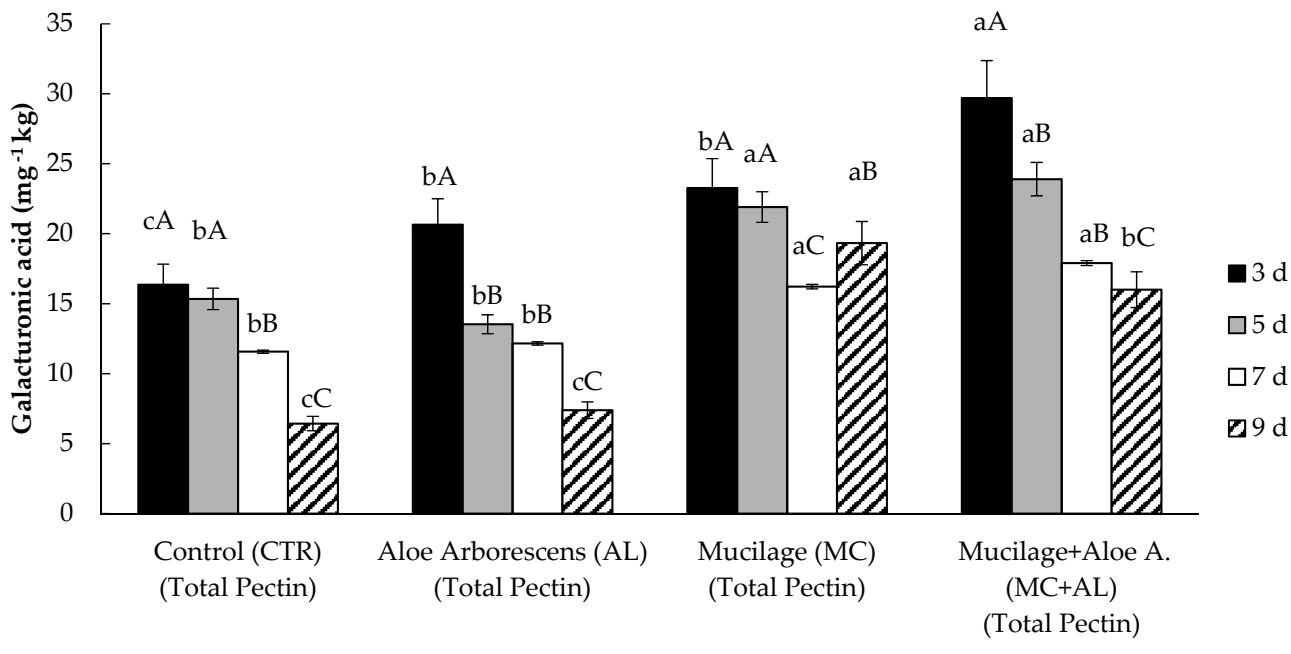

**Figure 4.** Total pectin (TP) of fruit slices of *Actinidia deliciosa* (Liang, Ferguson) coated with *O. ficus-indica* (OFI) mucilage (MC), or mucilage + *A. arborescens* (MC+AL), or *A. arborescens* (AL) not treated (CTR), just after being coated (0) and at 3, 5, 7, 9 days of storage at 5 °C. Different lowercase letters indicate significant differences between treatments at each sampling date. Different capital letters indicate significant differences between sampling date for the same treatment. Tukeys' significant test was applied at $p \leq 0.05$. Data are means ± S.E. ($n = 3$).

Pectin exists in the primary cell wall and the middle layer, which contribute to enhance the adhesion between cells and the mechanical strength of cells [38]. Kiwifruit slices coated with MC+AL and MC showed higher total pectin content than CTR and AL treatments for up to 9 d of storage (Figure 4). The pectin trends (Figures 4 and 5) influence the loss of firmness (Figure 2) of kiwifruit during storage, as reported by Allegra et al. [14]. Indeed, after cutting, the solubilization and depolymerization of the middle lamella of fruits might contribute to cell wall loosening and disaggregation [39]. During fruit storage, the middle lamella and primary cell wall structures were disassembled; meanwhile, pectin side chains depolymerized due to enzyme effects. These changes led to the loss of firmness, softening of fruit, and an increase of fruit postharvest decay, and decreased the quality of fresh fruit [40]. Therefore, the addition of OFI mucilage + *A. Arborescens* could significantly interfere with the assembly of structures by reducing the loss of cellular turgidity during

storage. Indeed, lack of membrane integrity allows the leakage of cellular osmotic solutes into the apoplastic space, which then results in water movements and turgor loss.

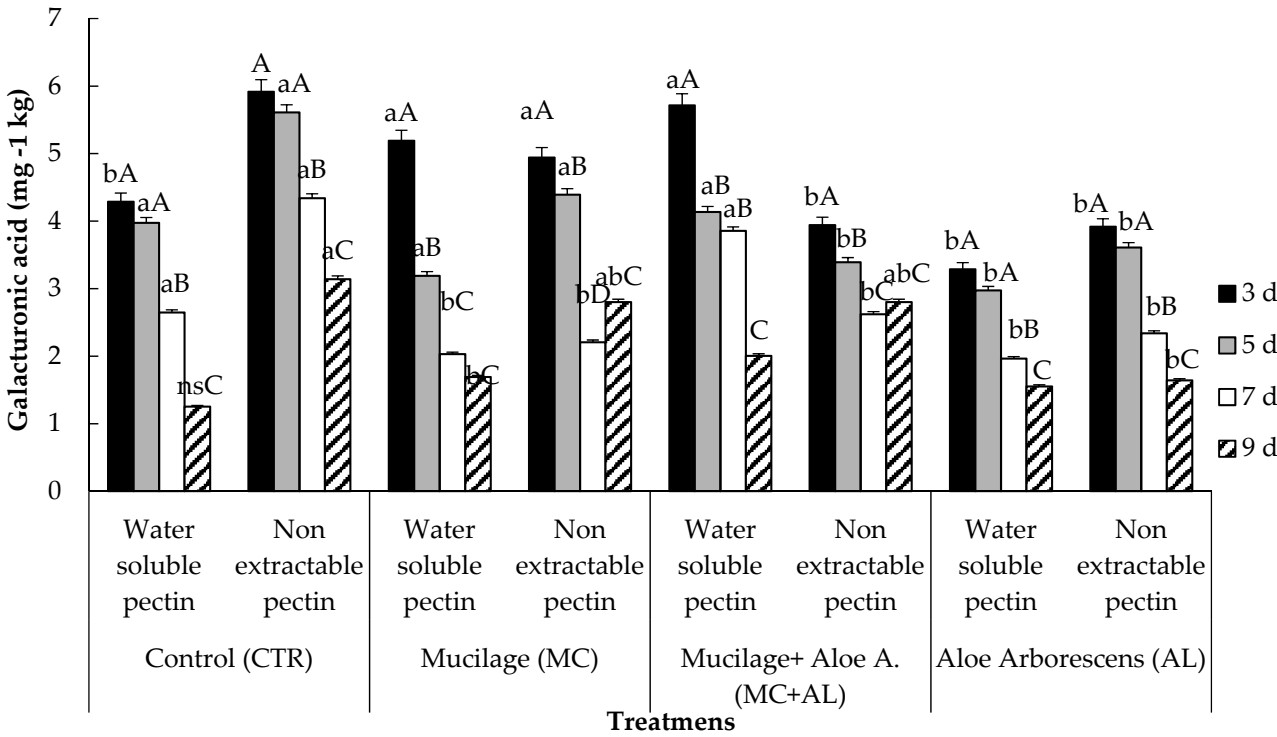

**Figure 5.** Water soluble pectin (WSP) and non-extractable pectin (NXP) of fruit slices of Actinidia deliciosa (Liang, Ferguson) coated with *O. ficus-indica* (OFI) mucilage + glycerol (MC), or OFI mucilage (MC), or mucilage + *A. arborescens* (MC+AL), or *A. arborescens* (AL) not treated (CTR), just after being coated (0) and at 3, 5, 7, 12 days of storage at 5 °C. Different lowercase letters indicate significant differences between treatments at each sampling date. Different capital letters indicate significant differences between sampling date for the same treatment. Tukeys' significant test was applied at $p \leq 0.05$. Data are means ± S.E. ($n = 3$).

Soluble pectin content in MC+AL and MC treatments showed a significant difference after cutting compared to CTR and AL treatment. At the 7th day of storage, kiwifruit coated with MC+AL showed a higher value than other treatments, but at the 9th day the values from the same treatment showed a reduction of 47.9% of soluble pectin content. Non-extractable pectin (Figure 5) significantly decreased in CTR and AL treatments during storage, while MC+AL and MC treatments after 7 d increased by 27.1 and 6.2%, respectively.

Similar results have been reported from Allegra e al. [14] on kiwifruit coated by mucilage extracted of *Opuntia ficus-indica*. Our results showed that the OFI effect influences the increase of pectin content during storage; on the contrary, when aloe coating has been used, a decrease of the pectin content is observed. In other works, *A. vera* and mucilage coating proved to be effective in reducing pectin depolymerization [3,26], while *A. arborescens* coating proved to reduce firmness loss in peach fruit during cold storage [16].

### 3.3. Head Spaces Gas Composition

Surface coatings can increase a fruit's skin resistance to gas diffusion, modify its internal atmosphere composition, and depress its respiration rate. In fresh fruit, the effect of covering the cuticle and blocking pores were important for resistance to water vapor diffusion [41]. However, the use of cover on skin could change internal atmosphere, respiration, and transpiration rates. This could be important in fresh-cut fruit because the use of coating substitute the removal of cuticle and pores that regulate changes of oxygen and carbon dioxide. In our work, after cutting and packaging in passive atmosphere,

kiwifruit showed a progressive significant increase of $CO_2$ and a significant decrease in $O_2$. The $O_2$ (Kpa) content inside packages showed significant differences between samples coated with MC and MC+AL compared to CTR and AL treatment after 3 d of storage. Untreated slices showed lower values than the MC+AL, AL, and MC samples (Figure 6) after 5 and 7 days. An increase of $CO_2$ production was observed after cutting in all sample coated and uncoated, but MC treatment showed the lowest values between MC+AL, AL treatments and CTR. At the 5th day of storage no significant difference occurred between all kiwifruit samples, while at the 7th day kiwifruit coated with MC+AL and CTR showed a significant difference compared to MC and AL treatments. After 7 d all samples showed $CO_2$ production (<15.8 Kpa). Our results showed that kiwi slices treated with the MC+AL solution had a lower $CO_2$ concentration in bags until 7 d. A similar result was observed by Benitez et al. [3] on kiwi slices coated with *A. vera* combined with package under passive atmosphere and low temperature ($2 \pm 1$ °C). The reduction of respiration was observed also in coated sweet cherry [42] and in table grapes [43].

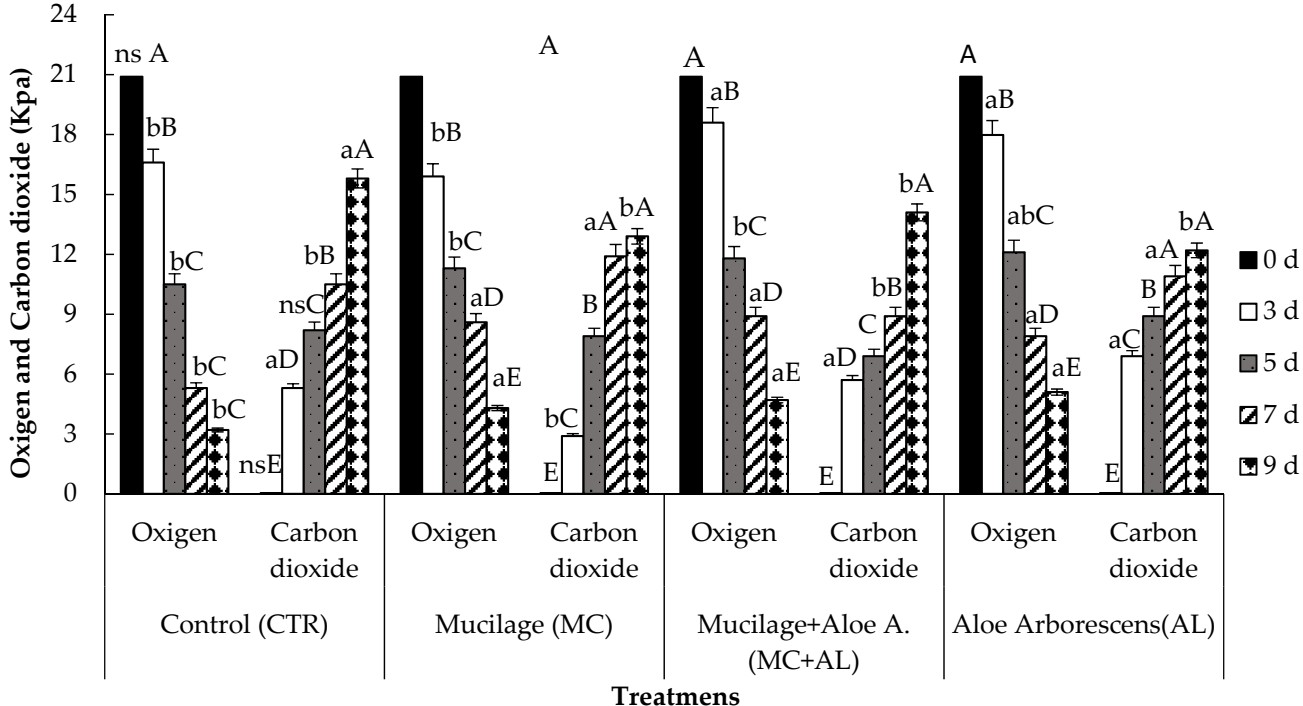

**Figure 6.** Oxygen and Carbon dioxide content (kPa) inside packages with fruit slices of Actinidia deliciosa (Liang, Ferguson) coated with *O. ficus-indica* (OFI) mucilage (MC), or mucilage + *A. arborescens* (MC+AL), or *A. arborescens* (AL) not treated (CTR), just after being coated (0) and at 3, 5, 7, 9 days of storage at 5 °C Different lowercase letters indicate significant differences between treatments at each sampling date. Different capital letters indicate significant differences between sampling date for the same treatment and parameter (Oxigen, carbon dioxide). Tukeys' significant test was applied at $p \leq 0.05$. Data are means $\pm$ S.E. ($n = 3$).

### 3.4. Microbiological Analysis

Aloe gel is made of a wide range of constituents that are mainly responsible for inhibiting the the growth of both Gram-positive and Gram-negative bacteria such as *Bacillus cereus*, *Salmonella typhimurium*, *Escherichia coli*, and *Klebsialla pneumonia* [44]. Allegra et al. [14] showed that the use of OFI mucilage on kiwifruit slices does not inhibit significantly the grown of yeast and bacteria; on the contrary, *A. arborescens* coating reduced the microbial spoilage of kiwifruit during 7 d of storage [3]. The loads of aerobic mesophilic bacteria, pseudomonas, and yeast detected on the different samples collected during the experimentations are reported in Table 3. The microbiological analyses of solutions of different coatings (MC, MC+AL, and AL) did not evidence the presence of any of the microbial

groups observed. The groups of aerobic mesophilic bacteria were detected after cutting and at the 3rd day of storage in all treatments with low concentration (<2 Log CFU g$^{-1}$). No differences occurred between MC+AL coated slices and AL at the 5th and 7th d of storage. At the same times, the values of CTR and AL showed an increase in this population, with a load of 3.2 and 3.6, and 4.2 and 5.4 Log CFU g$^{-1}$, respectively. The presence of pseudomonads were detected after 7 d of storage (Table 5).

**Table 5.** Total mesophilic, *Pseudomonads*, and yeast content (Log CFU g$^{-1}$) in fruit slices of *Actinidia deliciosa* (Liang. Ferguson) coated with *O. ficus-indica* (OFI) mucilage (MC), or mucilage + *A. arborescens* MC+AL, or *A. arborescens* (AL) not treated (CTR), just after being coated (0) and at 3, 5, 7, 9 days of storage at 5 °C. Different lowercase letters indicate significant differences between treatments at each sampling date. Different capital letters indicate significant differences between sampling date for the same treatment. Tukeys' significant test was applied at $p \leq 0.05$. Data are means ± S.E. ($n = 3$).

| Time of Storage (Days) | Treatments | | | | | | | | | | | |
|---|---|---|---|---|---|---|---|---|---|---|---|---|
| | Aerobic Mesophile Bacteria (Log CFU g$^{-1}$) | | | | Pseudomonads (Log CFU g$^{-1}$) | | | | Yeast Content (Log CFU g$^{-1}$) | | | |
| | CTR | MC | MC+AL | AL | CTR | MC | MC+AL | AL | CTR | MC | MC+AL | AL |
| 0 days | 0 nsD | 0 D | 0 B | 0 B | 0 nsC | 0 C | 0 C | 0 C | 0 nsC | 0 D | 0 C | 0 b |
| 3 days | 0.9 aC | 1.1 aC | 0 bB | 0 bB | 0 nsC | 0 C | 0 C | 0 C | 0 nsC | 0 D | 0 C | 0 B |
| 5 days | 1 bC | 2 aB | 0 cB | 0 cB | 0 nsC | 0 C | 0 C | 0 C | 0 cC | 2.9 aC | 0.1 bC | 0 cB |
| 7 days | 1.9 bB | 4.3 aA | 0 cB | 0 cB | 2 bB | 3.5 aB | 1 cB | 0.5 cB | 1 bB | 3.3 aB | 1.1 bB | 0 cB |
| 9 days | 3.3 bA | 5.4 aA | 1 cA | 1 cA | 4.5 aA | 5 aA | 1.5 bA | 1 bA | 2 bA | 4 aA | 1.5 bA | 1.3 cA |

Differences among treatments appeared after 7 and 9 d of storage when CTR, MC, and AL treatments were characterized by different concentrations of pseudomonads. No differences occurred between MC coated slices and CTR after 5 d of storage. Pseudomonad's content (Table 5) did not show significant differences in all samples during 5 d of storage, while a significant increase in this population occurred in MC, showing a load of 1.77 Log CFU g$^{-1}$ after 7 d. A similar trend was also observed for yeasts. In this case, the highest increase in concentration was registered on MC and MC+AL kiwifruit slices after 5 and 7 days. The addition of *A. arborescens* to the mucilage coating solution significantly reduced the aerobic mesophilic bacteria, pseudomonads, and yeast contents on kiwifruit slices during storage time. However, the cell densities of pseudomonads and yeasts were not able to determine a microbial decay of the sliced kiwifruits (Table 5).

## 4. Conclusions

This work reported the efficiency of *A. arborescens* + *Opuntia ficus-indica* coating on fresh-cut kiwi fruit. *A. arborescens* + *Opuntia ficus-indica* mucilage coating showed significant benefits in the postharvest quality of fresh-cut 'Hayward' kiwifruit, retarding fruit ripening and maintaining visual score and firmness. Moreover, it maintained the titratable acidity and total soluble solids and reduced the weight loss, in addition avoiding the microbial activity. *A. arborescens* + *Opuntia ficus-indica* coating mucilage allowed a gain of three days for commercial shelf life and 7 days for visual score parameter, as compared to the uncoated fresh-cut fruit.

**Author Contributions:** Conceptualization, G.S., A.A. and P.I.; methodology, G.S. and A.A.; validation, V.F., A.A. and G.S.; formal analysis, G.S., A.A. and R.P.; investigation, A.A., V.F. and G.S.; resources, A.A., V.F. and G.S.; software, A.A., R.P. and G.S.; data curation, V.F., A.A. and G.S.; writing—original draft preparation, A.A., P.I. and G.S.; writing—review and editing, G.S., A.A. and P.I.; visualization G.S. and A.A.; supervision, A.A., G.S. and P.I. All authors have read and agreed to the published version of the manuscript.

**Funding:** This research was funded by FFR2021 Giuseppe Sortino.

**Institutional Review Board Statement:** Not applicable.

**Informed Consent Statement:** Not applicable.

**Data Availability Statement:** All data generated or analyzed during this study are included in this published article.

**Conflicts of Interest:** The authors declare no conflict of interest.

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
