# Peer review of "The Use of Opuntia ficus-indica Mucilage and Aloe arborescens as Edible Coatings to Improve the Physical, Chemical, and Microbiological Properties of ‘Hayward’ Kiwifruit Slices"

_horticulturae, doi:10.3390/horticulturae8030219_

Round 1
Reviewer 1 Report
The coating method was adopted to preserve the fresh-cut kiwifruit slices using the natural mucilage of Opuntia ficus-indica or Aloe arborescens. The results indicated that Aloe arborescens had a positive effect on the quality change of kiwifruit slices during storage. The experimental results provide a theoretical basis for the application of fresh-cut kiwifruit preservation in practice. A systematic study was carried out on the effect of mucilage on the quality of fresh-cut kiwifruit. The experimental method description is also comprehensive.
However, there is a lack of innovation in the design of the experiment. The authors only focused on the changes in quality of fresh-cut kiwifruit slice. But, from what I understand similar results have been reported in previous studies. The overall goal of this article is on practical application. Except for the quality, the authors should pay more attention to the changes of nutritional composition and sensory of kiwifruit slice. For example, the kiwifruit is the king of Vc in fruit, which has strong antioxidant properties. Then, whether fresh-cut kiwifruit slices coating by mucilage make a negative effect in taste is critical to consumers.
The presentation of the experimental results needs to be further improved. Table 1 and Table 2 are more clearly presented in the form of a line graph. And the table is not a canonical three-line table.
The first paragraph of the introduction could be divided into several paragraph. It is important to clarify the significance of needing to coat fresh-cut kiwifruit slice with mucilage.
The analysis of the results should be more in-depth on the content of different types of pectin to explain the internal reasons for the softening of kiwifruit after slicing.
Line 231, what is the reason for the increase in SST only on day 3? The reason for the change in TA, including line 238, is not explained. It is reasonable to analyze the trend of organic acid composition, because this indicator has a great impact on the taste.
Line 308 mentions that compound polysaccharide coating further reduces weight loss and hardness, and the results of this study show that MC+AL has the highest hardness, which should give a reasonable explanation here (although it is mentioned later that the maximum hardness in the MC+AL treatment group may be due to the use of OFI mucilage with hydrophilic properties). Can the change in hardness here be explained in relation to the degree of spoilage as judged by coloration earlier?
Line 399 has incorrect CO2 formatting, line 393 has incorrect formatting, and line 390 has extra spaces.
3.3 When analyzing the gas in the package, it is not clear what effect the concentrated coating used for the experiment has on the kiwi slices in the end.
Line 469: This is a fatal mistake.
Author Response
Thank you for your suggestions; we improved our manuscript.
Kind regards,
Dr. Alessio Allegra
---------
-----------
The presentation of the experimental results needs to be further improved. Table 1 and Table 2 are more clearly presented in the form of a line graph. And the table is not a canonical three-line table.
AU: we have formatted Tables 1 and 2 as recommended by the guidelines for authors
The first paragraph of the introduction could be divided into several paragraph.
AU: we followed the recommendation given and We have improved the text
The analysis of the results should be more in-depth on the content of different types of pectin to explain the internal reasons for the softening of kiwifruit after slicing.
AU: We have improved the text to point 3.2 Firmness, weight loss and pectin content
Line 231, what is the reason for the increase in SST only on day 3? The reason for the change in TA, including line 238, is not explained. It is reasonable to analyze the trend of organic acid composition, because this indicator has a great impact on the taste.
AU: We have improved the text to point 3.1 Solid soluble total (SST), Titratable acidity (TA), Color and Visual score
Line 308 mentions that compound polysaccharide coating further reduces weight loss and hardness, and the results of this study show that MC+AL has the highest hardness, which should give a reasonable explanation here (although it is mentioned later that the maximum hardness in the MC+AL treatment group may be due to the use of OFI mucilage with hydrophilic properties). Can the change in hardness here be explained in relation to the degree of spoilage as judged by coloration earlier?
AU: We have improved the text to point We have improved the text to point 3.2 Firmness, weight loss and pectin content
Line 399 has incorrect CO2 formatting, line 393 has incorrect formatting, and line 390 has extra spaces.
AU: done
3.3 When analyzing the gas in the package, it is not clear what effect the concentrated coating used for the experiment has on the kiwi slices in the end.
AU: we have improved the text to point 3.3 Head spaces gas composition
Reviewer 2 Report
To whom it may concern;
The manuscript entitled “The use of Opuntia ficus-indica mucilage and Aloe arborescens as edible coating to improve physical, chemical and microbiological properties of ‘Hayward’ kiwifruit slices” presented results regarding postharvest application of Opuntia ficus-indica mucilage, Aloe arborescens and their combinations on fresh-cut kiwifruit “Hayward” to enhance storability through storage (Shelf-life storage actually). The current study reported interesting results still needs deep and strong attention and additionally considering comments to be published, in my opinion. It seems that the manuscript (MS) was written carelessly without enough attention to the rules of paper writing. Consequently, many writing problems and mistakes were recognized through the MS regarding grammatical points and words usages, as well. The MS must be read once more to solve problems. Results were not presented in good manner and need to be improved; discussion was poor and should be improved by using other works and studies and giving reasons, as well. In some parameters, no discussion was presented!! Conclusion section must be rewritten by giving the important results of the current study and final conclusion. More references could be used to improve the quality of the MS particularly discussion section. Legends of the tables and figures should be improved, as well. Most parts were highlighted through the MS.
Some comments are listed below:
Abstract:
Line21: “Showed” instead of shows
Note: better to have same pattern and time usage with other sentences that presented the results.
Line 23: in “the” slices
No conclusion was presented in the abstract!!! You should have at least one sentence considering conclusion….
*No enough words regarding keywords… better to have two more keywords.
Introduction:
Line39. A. arborescens
42: Italic
47,49: use A. arborescens and A. vera instead of Aloe arborescens and Aloe vera like line 43.
55: clarify OFJ in the first place and then use it.
*If you use or give an abbreviation for a word you should use it in the next usages.
M&M:
Line 66: you bought fruits from a supermarket!!!! In my opinion, it was not a true way since you had no idea about the harvesting date, any cold storage conditions before reaching supermarket or any other treatments, as well!! Any chilling injuries or microbial infections!!! These data could change all obtained data! Better to have fruits from orchard in next studies….
Why did fruits loss their firmness through a day? 48 to 40 N?
*Plz check the whole MS for English writing, Italic, abbreviations etc.
134,135,136: CO2, O2…… Correct
174: clarify AlS
244: Italic
253: , respectively
*In case of defining an abbreviation for a word or a treatment etc., there is an obligation to use that abbreviation in the next usages through the MS. Therefore, plz check the MS and correct all.
Results and Discussion:
Through results clarify if the changes you recorded were good or bad, for instance through lines 256-262
263: decrease of what?
263-267: why a new paragraph was made????
Color or colour???!! Plz use one form through the MS!
*Through results section give clear results of which treatment acted the best at each parameter? Why?
293: abbreviation
299: abbreviation or at least "control”
*No enough discussion was presented regarding most parameters. No enough comparison to other previous works and no enough reasons or possible reasons.
* no enough attention to the process of discussion
319: in all treatments
319: in the M&M section the firmness 48 reached to 40 N through a day before treatments. How was the same pattern observed for these treatments, as well?!
*In all results, no clear way of presenting presented. In fact, results were not clear and fluent.
337: space correction
342: treatments
351: correct……
352: through the MS you should use same word for instance day or day. or d or d. or days. or days
You could not use different forms for a word….. correct all!
352: correct verb
353: based on the journal policy no need for “ the year for reference”…. Correct
354: what was “the significant difference”? decrease or increase! Good or bad? Many results were presented in such ways that were not sufficient. Plz check all results parts and change the presenting way to have clear results.
357,358: grammatical mistakes….correct
Similar mistakes through the MS.
377: correct
378: plz use same pattern of writing through the MS.
378-380: use past form of verbs due to presenting the current results not facts!
380,381: Plz write same forms for mucilage (MC), A. arborescens, A. vera
389: CO2 and O2 instead of carbon….
393: d. ; d; day ; day. ; days ; days.
Correct as commented previously
394,397,398,399,402,417,418: correct patterns of writings for day or day. or d or d. or….and Co2 and O2 , treatments, etc.
422: grammatical mistakes
447: d. and why using control trails instead of control
448: treatments
449: grammar
447-458: check for the previous mentioned corrections
450: all samples
454: grammar
455: as previously mentioned how did it/they affect? Good or bad? Decrease or increase? In addition, correct grammar!
*The presenting way of the results seems to be insufficient, made it difficult to set clear observation….
*mostly no discussion is available for….
461: remove “the”
463-464: references were wrongly presented; additionally, no need of references generally in conclusion
465: Aleo A. ??????????????????????
465-467: rewrite sentences

Author Response
The manuscript entitled “The use of Opuntia ficus-indica mucilage and Aloe arborescens as edible coating to improve physical, chemical and microbiological properties of ‘Hayward’ kiwifruit slices” presented results regarding postharvest application of Opuntia ficus-indica mucilage, Aloe arborescens and their combinations on fresh-cut kiwifruit “Hayward” to enhance storability through storage (Shelf-life storage actually). The current study reported interesting results still needs deep and strong attention and additionally considering comments to be published, in my opinion. It seems that the manuscript (MS) was written carelessly without enough attention to the rules of paper writing. Consequently, many writing problems and mistakes were recognized through the MS regarding grammatical points and words usages, as well. The MS must be read once more to solve problems. Results were not presented in good manner and need to be improved; discussion was poor and should be improved by using other works and studies and giving reasons, as well. In some parameters, no discussion was presented!! Conclusion section must be rewritten by giving the important results of the current study and final conclusion. More references could be used to improve the quality of the MS particularly discussion section. Legends of the tables and figures should be improved, as well. Most parts were highlighted through the MS.
Some comments are listed below:
Abstract:
Line21: “Showed” instead of shows
Note: better to have same pattern and time usage with other sentences that presented the results.
AU: Done
Line 23: in “the” slices
AU: Done
No conclusion was presented in the abstract!!! You should have at least one sentence considering conclusion….
AU: We have improved the abstract as recommended
*No enough words regarding keywords… better to have two more keywords.
AU: Done
Introduction:
Line39. A. arborescens
AU: Done
42: Italic
AU: Done
47,49: use A. arborescens and A. vera instead of Aloe arborescens and Aloe vera like line 43.
AU: Done
55: clarify OFJ in the first place and then use it.
*If you use or give an abbreviation for a word you should use it in the next usages.
AU: Done
M&M:
Line 66: you bought fruits from a supermarket!!!! In my opinion, it was not a true way since you had no idea about the harvesting date, any cold storage conditions before reaching supermarket or any other treatments, as well!! Any chilling injuries or microbial infections!!! These data could change all obtained data! Better to have fruits from orchard in next studies….
AU: thank you for your consideration.
Why did fruits loss their firmness through a day? 48 to 40 N?
AU: sorry there is mistake
*Plz check the whole MS for English writing, Italic, abbreviations etc.
AU: Done
134,135,136: CO2, O2…… Correct
AU: Done
174: clarify AlS
AU: Done
244: Italic
AU: Done
253: , respectively
AU: Done
*In case of defining an abbreviation for a word or a treatment etc., there is an obligation to use that abbreviation in the next usages throug
AU: Done
h the MS. Therefore, plz check the MS and correct all.
AU: Done
Results and Discussion:
Through results clarify if the changes you recorded were good or bad, for instance through lines 256-262
AU: Done
263: decrease of what?
AU: there was a mistake in it.
263-267: why a new paragraph was made????
AU: is not a new paragraph
Color or colour???!! Plz use one form through the MS!
AU: Done
*Through results section give clear results of which treatment acted the best at each parameter? Why?
293: abbreviation
AU: Done
299: abbreviation or at least "control”
AU: Done
*No enough discussion was presented regarding most parameters. No enough comparison to other previous works and no enough reasons or possible reasons.
AU: We have improved the results and discussion
* no enough attention to the process of discussion
319: in all treatments
AU: Done
319: in the M&M section the firmness 48 reached to 40 N through a day before treatments. How was the same pattern observed for these treatments, as well?!
AU: there was a mistake in it.
*In all results, no clear way of presenting presented. In fact, results were not clear and fluent.
337: space correction
AU: Done
342: treatments
AU: Done
351: correct……
AU: Done
352: through the MS you should use same word for instance day or day. or d or d. or days. or days
You could not use different forms for a word….. correct all!
AU: Done
352: correct verb
AU: Done
353: based on the journal policy no need for “ the year for reference”…. Correct
AU: Done
354: what was “the significant difference”? decrease or increase! Good or bad? Many results were presented in such ways that were not sufficient. Plz check all results parts and change the presenting way to have clear results.
AU: Done
357,358: grammatical mistakes….correct
AU: Done
Similar mistakes through the MS.
AU: Done
377: correct
AU: Done
378: plz use same pattern of writing through the MS.
AU: Done
378-380: use past form of verbs due to presenting the current results not facts!
AU: Done
380,381: Plz write same forms for mucilage (MC), A. arborescens, A. vera
AU: Done
389: CO2 and O2 instead of carbon….
AU: Done
393: d. ; d; day ; day. ; days ; days.
Correct as commented previously
AU: Done
394,397,398,399,402,417,418: correct patterns of writings for day or day. or d or d. or….and Co2 and O2 , treatments, etc.
AU: Done
422: grammatical mistakes
AU: Done
447: d. and why using control trails instead of control
AU: Done
448: treatments
AU: Done
449: grammar
AU: Done
447-458: check for the previous mentioned corrections
AU: Done
450: all samples
AU: Done
454: grammar
AU: Done
455: as previously mentioned how did it/they affect? Good or bad? Decrease or increase? In addition, correct grammar!
AU: We have correct the text
*The presenting way of the results seems to be insufficient, made it difficult to set clear observation….
*mostly no discussion is available for….
AU: We have improved the results and discussion
461: remove “the”
AU: Done
463-464: references were wrongly presented; additionally, no need of references generally in conclusion
AU: We have rewrite the conclusion
465: Aleo A. ??????????????????????
AU: Done
465-467: rewrite sentences
AU: We have rewrite the conclusion
Thank you for your suggestions; we improved our manuscript.
Kind regards,
Dr. Alessio Allegra
Reviewer 3 Report
Manuscript ID: horticulturae-1605265
Type of manuscript: Article
Title: The use of Opuntia ficus-indica mucilage and Aloe arborescens as edible coating to improve physical, chemical and microbiological properties of ‘Hayward’ kiwifruit slices
The manuscript evaluate the efficacy of O. ficus-indica coat ings with Aloe arborescens to prolong the shelf life of kiwifruit slices during 9 days of storage. The manuscript required corrections. I hope that the following suggestions and comments will be helpful.
The authors should take full responsibility for typo and grammatical errors existing in this manuscript. Also, some additional information in the methodology part is required. The statistics have to be corrected. The conclusion part required rewriting.
My detailed comments on the article are listed directly in the pdf file.

Author Response
Dear Reviewer 3,
Thank you for your suggestions; we improved our manuscript.
We have answered all recommendations
Kind regards,
Dr. Alessio Allegra

Round 2
Reviewer 1 Report
The author still needs to enhance the English language and correct the details of the experiment.
Author Response
Dear Reviewer 1,
Thank you for your suggestions; we improved our manuscript.
We have answered all recommendations of reviewer 2 and 3
Kind regards,
Dr. Alessio Allegra
Reviewer 2 Report
some points were commented in the MS.
Please check all the MS for mistakes. I noticed some points that need to be improved.

Author Response
Dear Reviewer 2,
Thank you for your suggestions; we improved our manuscript.
We have answered all recommendations
Kind regards,
Dr. Alessio Allegra

Reviewer 3 Report
The authors corrected the article, however, still some corrections are needed. There is a lot of editing errors - please correct them in the manuscript (I did not mark them all - please carefully review the article and correct it). More comments you will find in the attached document.

Author Response

(The authors gave the same response as above.)
